# Synthesis and Evaluation of Dodecaboranethiol Containing Kojic Acid (KA-BSH) as a Novel Agent for Boron Neutron Capture Therapy

**DOI:** 10.3390/cells9061551

**Published:** 2020-06-25

**Authors:** Koji Takeuchi, Yoshihide Hattori, Shinji Kawabata, Gen Futamura, Ryo Hiramatsu, Masahiko Wanibuchi, Hiroki Tanaka, Shin-ichiro Masunaga, Koji Ono, Shin-Ichi Miyatake, Mitsunori Kirihata

**Affiliations:** 1Department of Neurosurgery, Osaka Medical College, 2-7 Daigaku-machi, Takatsuki-shi, Osaka 569-8686, Japan; neu133@osaka-med.ac.jp (K.T.); neu130@osaka-med.ac.jp (G.F.); neu106@osaka-med.ac.jp (R.H.); neu-wanibuchi@osaka-med.ac.jp (M.W.); 2Research Center of Boron Neutron Capture Therapy, Osaka Prefecture University, 1-1 Gakuen-cho, Nakaku, Sakai 599-8531, Japan; y0shi_hattori@riast.osakafu-u.ac.jp (Y.H.); kirihata@biochem.osakafu-u.ac.jp (M.K.); 3Institute for Integrated Radiation and Nuclear Science, Kyoto University, 2-1010 Asashiro-Nishi, Kumatori-cho, Sennan-gun, Osaka 590-0494, Japan; h-tanaka@rri.kyoto-u.ac.jp (H.T.); masunaga.shinichiro.6m@kyoto-u.ac.jp (S.M.); 4Kansai BNCT Medical Center, Osaka Medical College, 2-7 Daigakumachi, Takatsuki-shi, Osaka 569-8686, Japan; ono.koji.48x@st.kyoto-u.ac.jp (K.O.); neu070@osaka-med.ac.jp (S.-I.M.)

**Keywords:** boron neutron capture therapy (BNCT), F98 rat glioma model, kojic acid, boron compound

## Abstract

Boron neutron capture therapy (BNCT) is a form of tumor-cell selective particle irradiation using low-energy neutron irradiation of boron-10 (^10^B) to produce high-linear energy transfer (LET) alpha particles and recoiling ^7^Li nuclei (^10^B [n, alpha] ^7^Li) in tumor cells. Therefore, it is important to achieve the selective delivery of large amounts of ^10^B to tumor cells, with only small amounts of ^10^B to normal tissues. To develop practical materials utilizing ^10^B carriers, we designed and synthesized novel dodecaboranethiol (BSH)-containing kojic acid (KA-BSH). In the present study, we evaluated the effects of this novel ^10^B carrier on cytotoxicity, ^10^B concentrations in F98 rat glioma cells, and micro-distribution of KA-BSH in vitro. Furthermore, biodistribution studies were performed in a rat brain tumor model. The tumor boron concentrations showed the highest concentrations at 1 h after the termination of administration. Based on these results, neutron irradiation was evaluated at the Kyoto University Research Reactor Institute (KURRI) with KA-BSH. Median survival times (MSTs) of untreated and irradiated control rats were 29.5 and 30.5 days, respectively, while animals that received KA-BSH, followed by neutron irradiation, had an MST of 36.0 days (*p* = 0.0027, 0.0053). Based on these findings, further studies are warranted in using KA-BSH as a new B compound for malignant glioma.

## 1. Introduction

Malignant gliomas are the most common malignant primary brain tumors in adults [1]. These tumors are resistant to conventional therapy, including surgery, chemotherapy, and radiation therapy. One of the reasons for this is that these tumor cells invade the adjacent normal brain tissue [2]. Despite numerous clinical trials over the past 30 years, the prognosis for malignant gliomas, especially glioblastoma, is extremely poor [3,4]. To develop a treatment modality that can selectively target tumor cells without excessive damage to the normal tissue, we focused on boron neutron capture therapy (BNCT).

BNCT is based on the nuclear capture and fission reactions that occur when nonradioactive boron-10 (^10^B) is irradiated with neutrons of the appropriate energy to yield high linear energy transfer (LET) alpha particles (^4^He) and recoiling lithium-7 (^7^Li) nuclei. Because these particles have path lengths of approximately 5–9 µm, which is one cell diameter, the destructive effect is limited to ^10^B-containing cells. Therefore, it is important to achieve the selective delivery of large amounts of ^10^B to tumor cells with only small amounts of ^10^B to normal tissues.

A wide variety of boron delivery agents, such as amino acids, liposomes, and porphyrins have been synthesized and evaluated [5,6,7,8,9,10,11,12], but only two compounds are being used in clinical trials. The first, which has been used primarily in Japan, is dodecaboranethiol (BSH), and the second is the dihydroxyboryl derivative of phenylalanine, boronophenylalanine (BPA). These clinical trials showed a good therapeutic effect, but their overall results were not satisfactory [13,14,15].

BSH and its derivatives are of increasing interest as boron carriers for BNCT due to their potential to deliver large quantities of ^10^B atoms to tumor cells [16]. BSH is part of a class of water-soluble B cluster compounds and has low toxicity compared to other B cluster compounds. Owing to these properties, BSH is clinically used as the ^10^B carrier for BNCT of brain tumors with [17], although tumor selectivity of BSH is slightly low. To develop practical materials utilizing ^10^B carriers, we previously designed, synthesized, and evaluated various BSH-containing organic compounds, which constitute a new class of tumor-seeking and water-soluble compounds, such as amino acids, peptides, etc. The in vitro evaluation of cytotoxicity, cell-killing effects, and micro-distribution analysis by immunocytochemical technique suggested that BSH-containing, tumor-seeking compounds might be potential lead compounds for developing novel B compounds for BNCT [18,19,20].

Alternatively, kojic acid (5-hydroxy-4-pyran-4-one-2-methyl, KA) is obtained from Kouji (fungi native to Japan) and is a byproduct of the fermentation process for the alcoholic beverage Japanese sake and soy sauce [21]. KA was approved in 1989, and has been widely used in skincare products, especially for lightening skin pigmentation because KA has an inhibition activity for melanogenesis process of tyrosinase [22]. Furthermore, KA and its derivatives exhibit various biological activities, such as antioxidative, radioprotective, antiproliferative, and anti-inflammatory effects [23,24,25]. Therefore, we were motivated to develop a novel B carrier for BNCT that contains KA as the tumor-seeking moiety.

In this study, we designed and synthesized the novel BSH-containing, tumor-seeking compound KA-BSH as a ^10^B carrier for BNCT. Furthermore, the in vitro and in vivo evaluation of KA-BSH is also described.

## 2. Materials and Methods

### 2.1. Synthesis

#### 2.1.1. General

Brominated KA (3) was prepared according to the method by Bertrand et al. (2004) [26]. *S*-cyanoethylthio-dodecaborate (2) was prepared according to the method described previously by Hattori et al. (2011) [27]. Flash ODS column chromatography was performed using Isolera Spektra (Biotage Sweden AB, Uppsala, Sweden) with a SNAP KP-C18-HS Cartridge (Biotage Sweden AB, Uppsala, Sweden). ^1^H-NMR spectra were measured on a JMTC-400/54/SS (400 MHz, JEOL Ltd., Tokyo, Japan). ESI-MS measurements were performed on an EXACTIVE (Thermo Fisher Scientific, Waltham, MA, USA).

#### 2.1.2. Synthesis of S-(2-(5-hydroxy-4-oxo-4H-pyran-2-yl) methyl) Thioundecahydro-Closo- Dodecaborate (2-) Disodium Salt (KA-BSH, 1)

To a solution of compound 3 (202 mg, 0.99 mmol) in MeCN (20 mL) was added *S*-cyanoethylthio-dodecaborate (2) (300 mg, 0.82 mmol), and the mixture was refluxed for 24 h. The reaction mixture was concentrated, and the resulting residue was suspended in acetone (50 mL). After filtration, 25 *w*% NMe_4_OH in MeOH (354 mg, 0.97 mmol) was added to the filtered solution. The generated white solid was filtrated and washed with acetone, and the residual solid was dissolved in H_2_O. The solution was acidified with Amberlite IR-120(H^+^) to remove tetramethyl ammonium salt. The mixture filtered and the solution was neutralized with 1 N NaOH. The mixture was purified by flash ODS column chromatography to give 1 (219 mg, 80%) as a colorless powder. ^1^H-NMR(D_2_O): δ = 0.75–1.80 (11H, mbr), 3.50 (2H, s), 6.47 (s, 1H), 7.20 (s, 1H). ^13^C-NMR(D_2_O): δ = 33.85, 112.81, 142.68, 145.04, 170.62, 177.39. MS: (neg. ESI, m/z) 144.1189 [A]^2-^.

### 2.2. Boron Compounds

BPA (L-isomer), which is currently used in clinical BNCT studies, was provided by Stella Pharma Corporation (Osaka, Japan) and was prepared as a D-fructose complex. BSH was purchased from Katchem Ltd. (Katchem, Prague, Czech Republic). KA-BSH was prepared as described above.

### 2.3. Cell Culture

F98 rat glioma, C6 rat glioma, and B16 mouse melanoma cell lines, used in vitro boron uptake experiments. F98 rat glioma cells produce infiltrating tumors in the brains of Fischer rats. The tumors are refractory to chemotherapy and radiotherapy. F98 rat glioma cells have been characterized as anaplastic or undifferentiated glioma [28]. F98 rat glioma cells were provided by Dr. Barth (Department of Pathology, the Ohio State University, Columbus, OH). These cells were cultivated in a Dulbecco’s modified Eagle medium (DMEM) containing 10% fetal bovine serum (FBS) and antibiotics (penicillin and streptomycin) at 37°C in an atmosphere of 5% CO_2_. These materials for the culture medium were purchased from Gibco Invitrogen (Grand Island, NY, USA).

### 2.4. F98 Rat Glioma Model.

All animal experimental procedures were approved by the Animal Use Review Board and Ethical Committee of Osaka Medical College (No. 25092, 28 October 2015) and Kyoto University Research Reactor Institute (KURRI, No. 25030-13, 1 April 2015), and performed following the Guide for the Care and Use of Laboratory Animals. Eight-week-old, male Fischer 344 rats weighing between 200 and 250 g were used (Japan SLC, Inc. Shizuoka, Japan). Rats were anesthetized with an intraperitoneal (ip) injection of a mixture of medetomidine 0.15 mg, midazolam 2 mg, and butorphanol 2.5 mg/kg body weight (bw). The head of each rat was fixed with a stereotactic frame (Model 900; David Kopf Instruments, Tujunga, CA, USA). The scalp was incised at the midline and a small burr hole was drilled into the skull (1 mm posterior and 4 mm right-lateral to the bregma), using an electric drill. And then, F98 glioma cells were injected into the brain using a 25 µL Hamilton microsyringe with a 26-gauge needle (model 1700 RN, Hamilton Bonaduz AG, Bonaduz, Switzerland). The needle was inserted at a 5 mm depth from the skull and F98 cells were diluted in 10 µL of DMEM containing low-gelling temperature agarose at a concentration of either 10^3^ for therapeutic experiments or 10^5^ cells for biodistribution experiments. They were injected over 30 s using an automatic infuser pump. The needle entry hole was immediately sealed with bone wax and the incised scalp was closed with 3-0 silk sutures (Ethicon, North Ryde, NSW). After surgery, all animals were given atipamezole hydrochloride (0.15 mg/kg bw, ip).

### 2.5. In Vitro Uptake Experiments in F98 Cells

For the in vitro ^10^B uptake study, B16 mouse melanoma cells, C6 rat glioma cells, and F98 rat glioma cells were used. B16 and C6 cells were used for in vitro testing at Osaka Prefecture University. Based on results from initial in vitro screening, the in vitro ^10^B uptake study using F98 glioma cells were performed at Osaka Medical College. Four hundred thousand cells were seeded in a tissue culture dish (100 × 20 mm; Becton Dickinson, Franklin Lakes, NJ, USA) with DMEM supplemented with 10% FBS and 10% penicillin and streptomycin at 37 °C in a 5% CO_2_ atmosphere. After incubation for 4 days at 37 °C, the medium was removed, and DMEM with 10% FBS containing 1 mM KA-BSH, BSH, or BPA was added to each dish. The cells were incubated for an additional 24 h at 37 °C. The medium was then removed, and the cells were washed twice with 4 °C phosphate-buffered saline (PBS) and detached with trypsin–ethylenediamine tetraacetic acid solution. PBS was then added, and cells were digested overnight with 1 N nitric acid solution (Wako Pure Chemical Industries, Osaka, Japan), and ^10^B uptake was determined by inductively coupled plasma atomic emission spectrometry (ICP-AES) using an iCAP6000 emission spectrometer (Hitachi High-Technologies, Tokyo, Japan).

### 2.6. Immunostaining of F98 Cells

Immunostaining was performed to determine the incorporation of KA-BSH, BSH, or BPA into F98 cells, according to a previously described method with some modifications. Thereby, 35 mm glass-based dishes (IWAKI; Asahi Techno Glass, Shizuoka, Japan) were seeded with F98 cells (10^5^ cells suspended in 1 mL of DMEM with 10% FBS containing KA-BSH, BSH, or BPA). The final concentration was 2.0 mM in each case, and the cells were incubated for 24 h at 37 °C. Cells were treated with Hoechst 33342 at 37 °C for 30 min, after washing with DMEM, the cells were fixed with 10% paraformaldehyde in PBS for 10 min at room temperature. The cells were rinsed with PBS and then treated with 0.05% Triton X-100 for 10 min at room temperature. The cells were washed with PBS and incubated in a humid chamber with 1.0% BSA/0.02% NaN_3_ in PBS at room temperature for 60 min. After removing the solution, the cells incubated with the anti-BSH monoclonal antibody A9H3 (KA-BSH, BSH, and negative control groups) or the anti-BPA monoclonal antibody 2B10 (BPA group) in PBS containing 1.0% BSA/0.02% NaN_3_ (0.2 μg/mL) for 60 min at 32 °C. The cells were rinsed with PBS (3 times), and then incubated with Alexa-Fluor 488 goat anti-mouse IgG in PBS containing 1.0% BSA/0.02% NaN_3_ (KA-BSH, BSH, and control group: 0.2 μg/mL or BPA group: 0.5 μg/mL) for 30 min at 32 °C. After washing with PBS (3 times), the cells were mounted with Permafluor and then photographed with a microscope (IX-70, Olympus, Tokyo) equipped with a cooled, charge-coupled device camera (UIC-QE, Molecular Devices Co., Sunnyvale, CA, USA) controlled by MetaMorph software (Molecular Devices Co.).

### 2.7. Cell Viability Assay in F98 Cells

The cytotoxicity of KA-BSH was assayed using the water soluble tetrazolium (WST)-8 assay with a cell-counting kit (Wako Pure Chemicals, Osaka, Japan) following a modified version of the manufacturer-provided protocols [20]. Relative cell survival was calculated according to Equation (1):(1)$$\mathrm{Relative}\;\mathrm{cell}\;\mathrm{survival}\;(\%)=\frac{\mathrm{absorbance}\;\mathrm{value}\;\mathrm{of}\;\mathrm{compound}\;\mathrm{treated}\;\mathrm{well}}{\mathrm{absorbance}\;\mathrm{value}\;\mathrm{of}\;\mathrm{untreated}\;\mathrm{well}}\times 100$$

### 2.8. Biodistribution of KA-BSH in F98 Glioma Bearing Rats

Biodistribution studies were performed in Fischer rats bearing intracerebral implants of F98 rat glioma cells. 10–13 days after tumor implantation of 10^5^ F98 glioma cells, when signs of a progressively growing tumor were evident (weight loss, lethargy, hunching, and ataxia), biodistribution studies were initiated at which time the animals had tumors weighing 90–180 mg. KA-BSH was administered by intravenous (iv) injection in three doses (10, 20, and 30 µg ^10^B/kg bw). To compare its biodistribution with that of BPA or BSH, another group of rats received an iv injection of BPA at a concentration of 10 mg ^10^B/kg bw or BSH at a concentration of 30 mg ^10^B/kg bw. Biodistribution was determined 1 and 3 h after termination of iv injection. Animals were euthanized by an overdose of sevoflurane following which tumors and normal tissues, consisting of ipsilateral brain, contralateral brain, blood, heart, lung, liver, spleen, kidney, skin, and muscle, were removed and weighed for ^10^B determination by ICP-AES.

### 2.9. In Vivo Therapeutic Experiments

Therapeutic experiments for BNCT were performed 14 days after stereotactic implantation of 10^3^ F98 glioma cells. Rats were transported to the Nuclear Reactor Laboratory at KURRI. The rats were then randomized based on weight into experimental groups of 4–6 animals as follows: Group 1, KA-BSH administered iv and via neutron irradiation; Group 2, neutron irradiation only; Group 3, untreated control. All irradiated rats were anesthetized with a mixture of medetomidine (0.15 mg/kg bw), midazolam (2 mg/kg bw) and butorphanol (2.5 mg/kg bw). BNCT was initiated 1 h after termination of iv administration of KA-BSH (10 mg ^10^B/kg bw). The rats were irradiated at a reactor power of 1 MW using the Heavy Water Neutron Irradiation Facility for 1 h. The therapeutic effect of BNCT was evaluated by the survival time of the animals. Radiation dose of BNCT experiments have been estimated using the same method as we have reported [11,29,30,31,32,33]. The equivalent dose for KA-BSH cannot be estimated because the CBEB for KA-BSH is unknown.2.10. Statistical analysis

The means and standard deviations (SD) were computed for ^10^B concentrations of F98 glioma cells and tissues of Fischer rats bearing intracerebral implants of the F98 glioma. Logistic regression was used to determine the significance of differences in cell viability. Statistical differences between means were calculated using the Student’s unpaired t-test. In therapeutic experiments, the median survival time (MST) was calculated for each group using the Kaplan–Meier estimate. Kaplan–Meier curves were also plotted for all groups. An overall log-rank test was performed to test for equality of survival curves between groups. Statistical analyses were performed using the software JMP 12 (SAS Institute Inc., Cary, NC, USA).

## 3. Results

### 3.1. Synthesis of KA-BSH

The syntheses of KA-BSH 2Na (1) were carried out as shown in Figure 1A. Condensation of brominated KA (3) [26], which is derived from KA with *S*-cyanoethylthiododecaborate (2) [34], and subsequent deprotection were performed according to a previously reported method for producing KA-BSH tetramethylammonium salt [18]. KA-BSH2NMe_4_ was treated with amberlite IR-120(H^+^) and subsequently neutralized with 1 N NaOH (aq) to yield KA-BSH 2Na (1).

### 3.2. In Vitro Uptake Experiments in Cancer Cells.

The boron concentrations of B16 mouse melanoma cells and C6 rat glioma cells are shown in Appendix A. KA-BSH was taken up into cancer cells. The boron concentrations of F98 rat glioma cells are shown in Figure 1. In particular, the Boron concentrations of F98 cells in the KA-BSH group were significantly higher than after exposure to BSH (0.821 ± 0.047 vs. 0.508 ± 0.072 µg ^10^B/10^7^ cells; *p* = 0.0008) and BPA (0.821 ± 0.047 vs. 0.654 ± 0.062 µg ^10^B/10^7^ cells; *p* = 0.0157). The ^10^B concentration of F98 cells 24 h after exposure to KA-BSH was the highest.

### 3.3. Immunostaining of F98 Cells

The results showed that the distribution of KA-BSH was different from that of BPA or BSH. BPA was reported as widely distributed in the cytoplasm and the cell nuclei with no regions in which the concentration of BPA is especially high [35,36]. Indeed, BPA was widely distributed in the cytoplasm and cell nuclei herein (Figure 2E–H). However, BSH did not pass through the cell membrane (Figure 2I–L). In contrast, KA-BSH was incorporated into the cell membrane of the F98 cells and aggregated on the fringe of the cell nuclei (Figure 2M–P).

### 3.4. Cell Viability Assay in F98 Cells

The cytotoxicity of the KA-BSH, BSH, or BPA toward F98 rat glioma cells was determined using the WST-8 test. IC_50_ values for KA-BSH, BSH, and BPA were 10.4, 1.68, and >20 mM, respectively. The cytotoxicity of KA-BSH was very low. However, the cytotoxicity of KA-BSH was higher than BPA.

### 3.5. Biodistribution of KA-BSH in F98 Glioma Bearing Rats

The biodistribution data for KA-BSH, BSH, and BPA administered iv is shown in Figure 3. The B concentrations in the tumor, selected normal tissues of F98 glioma bearing rats as well as the tumor-to-normal brain (T/Br) ratios are also shown in Figure 3. The highest ^10^B concentrations were in tumors of the KA-BSH group at 1 h after termination of administration. However, the ^10^B concentrations in tumors of the 10 mg ^10^B/kg bw KA-BSH/iv group were much lower than the 10 mg ^10^B/kg bw BPA/iv group at 1 h after termination of administration (1.42 ± 0.28 vs. 13.54 ± 4.82 μg ^10^B/g; *p* = 0.0003). The ^10^B concentration in tumors of the KA-BSH/iv 30 mg ^10^B/kg bw group was much higher than the BSH/iv 30 mg ^10^B/kg bw group (6.65 ± 0.25 vs. 5.49 ± 0.25 μg; *p* = 0.0095). T/Br ratios of the KA-BSH/iv 30 mg ^10^B/kg bw group, BSH/iv 30 mg ^10^B/kg bw group, and BPA/iv 10 mg ^10^B/kg bw group at 1 h after termination of administration were 15.27, 17.61, and 4.21, respectively (Appendix A).

^10^B concentrations in the kidney and liver at 1 h after administration of KA-BSH (30 mg ^10^B/kg) were 28.15 ± 7.96 and 10.33 ± 2.61 µg/g, respectively, and ^10^B concentrations at 3 h after administration were 5.27 ± 0.63 and 4.19 ± 1.79 µg/g, respectively. The ^10^B concentrations in blood at 1 and 3 h were 14.21 ± 3.17 and 2.35 ± 1.37 µg/mL, respectively. Based on these data, KA-BSH showed rapid clearance from normal tissues. Furthermore, there was a small amount of ^10^B accumulation in other organs, such as the spleen, heart, lung, muscle, and skin (Table A1).

### 3.6. In Vivo Therapeutic Experiments

The estimated physical radiation doses delivered to the tumor, brain, and blood were calculated according to the ^10^B concentrations summarized in Table 1. The physical radiation doses delivered to the tumor were 1.1 Gy for KA-BSH administered by iv administration. The corresponding normal brain doses were 1.0 Gy. The survival data of Group 1, 2, and 3 following BNCT showed that the MSTs after implantation were 35.4 ± 8.0, 30.2 ± 2.2, and 28.5 ± 3.1 days, respectively. The survival time of the rats’ group using KA-BSH (Group 1) was significantly higher compared to the neutron irradiation only group (Group 2) (*p* = 0.0053) (Figure 4 and Appendix A).

## 4. Discussion

In malignant brain tumors, such as malignant glioma, tumor cells grow and infiltrate adjacent normal brain tissue. Therefore, BNCT, which is a tumor cell selective particle radiation therapy, could be the ideal treatment for malignant glioma [37].

In our institute, we have clinically applied BNCT since 2002 as an essential adjuvant therapy for patients with malignant gliomas or malignant meningiomas using BPA and BSH in combination or BPA only. We have achieved superior outcomes for newly diagnosed glioblastoma, compared to those with standard treatment using nonselective X-ray irradiation therapy [14,15]. Additionally, we reported the effectiveness of re-irradiation using BNCT for recurrent cases with malignant glioma and malignant meningioma with acceptable toxicity [13,38]. However, these results using BPA or BSH were not satisfactory. Therefore, the challenge for the next generation of BNCT is to develop more effective B compounds for BNCT. In the past, various types of B compounds were designed, synthesized, and evaluated. However, alternatives to BPA and BSH are not yet available.

For BNCT to be successful, the B compounds must have the following properties: (1) high tumor-targeting selectively (T/N ratio >3–4:1), (2) low or no toxicity itself, (3) concentrations >20 µg ^10^B/g in tumor tissues, and (4) high water-solubility. KA is a tyrosinase inhibitor derived from various fungal species, such as *Aspergillus* and *Penicillium* [39,40]. It is also reported that KA derivatives may be useful as cancer target drugs. We designed, synthesized, and evaluated BSH-containing, tumor-seeking compounds with KA. The cytotoxicity of KA-BSH is very low compared to BSH due to high water solubility and extremely low permeability of cell membrane. In addition, it is expected to be a promising compound that can be safely administered to living organisms because KA-BSH showed rapid clearance from normal tissues. Because KA is involved in melanin biosynthesis, it was proposed as novel compound for melanoma-BNCT treatment as well as BPA. For this reason, we started to examine KA-BSH for its potential adaptation to gliomas. In the present study, KA-BSH was found to be a promising compound capable of delivering high concentrations of ^10^B to various tumors. The toxicity of KA-BSH to the normal brain has not been properly assessed, however an evaluation of this drug will definitely be necessary for clinical applications in the future.

The use of the novel KA-BSH compound in BNCT was effective in prolonging median survival time in a rat malignant glioma model, despite low tumor distribution. The therapeutic effect of BNCT using intravenous administration of KA-BSH was higher than expected from tumor ^10^B concentrations and physical radiation doses. According to the results, KA-BSH might have extremely high Compound Biological Effectiveness (CBE); an important index of treatment planning of BNCT [32]. One explanation may be the tendency for higher intracellular accumulation of KA-BSH. Immunostaining showed that KA-BSH was aggregated on the fringe of the cell nuclei.

KA-BSH has lower ^10^B concentrations in tumors than that of BPA, however KA-BSH has a higher tumor- targeting selectively and it is a safer drug that have less impact on normal tissues. In addition, the tumor accumulation is higher than that of BSH and we expect novel KA-BSH to serve as the improved versions of BSH.

In the present study, KA-BSH has a good therapeutic effect with a single iv administration although most other novel B compounds are unlikely to show any therapeutic effect other than using the convection-enhanced delivery which is an invasive drug delivery technique. Immunostaining of F98 cells showed different intracellular distribution for each compound; therefore, it can be expected that the use of KA-BSH in combination with BPA may have a further curative effect. There are several studies on the combined effect of BSH and BPA, these studies showed that the effect of BNCT was improved by the combination of these two B compounds [41,42]. We also reported that the tumor ^10^B concentration was increased by using BSH and BPA in combination [43]. In the present study, we only used cell lines of rat- or mouse origin. We would like to expand our experiments to use human cell lines in the near future.

To evaluate tissue distribution of BPA, 4-Borono-2-[^18^F] fluoro-phenylalanine ([^18^F] BPA) positron emission tomography (PET) was employed by Ishiwata et al., and Imahori et al. applied [^18^F] BPA-PET for the first clinical use of BNCT [44,45]. We also developed [^18^F] BPA-PET for planning treatment with BNCT using BPA. Then, we reported that [^18^F] BPA-PET is useful for differentiating radiation necrosis from true tumor progression [46]. In BNCT, [^18^F] BPA-PET is now indispensable for the identification of adaptation, treatment planning, dose assessment, determination of therapeutic effect, and identification of recurrence. However, for BSH-containing compounds, including KA-BSH, there are currently no available compounds labeled with a positron. Therefore, for clinical use, it is necessary to develop an evaluation system for compound distribution, such as fluoride-labeled KA ([^18^F] KA) PET.

Conventionally, a large-scale nuclear reactor facility is required to obtain the neutron beam, but recently, BNCT using accelerator-based neutron sources has been developed. Consequently, BNCT is becoming a common treatment for malignant tumors. However, the use of an accelerator is limited to clinical research. For nuclear reactors, the development of new compounds will be delayed due to restrictions on available machine time among other reasons. Therefore, it is desirable to have additional accelerators installed to promote research and development for BNCT. The development of novel B compounds with high tumor accumulation selectively is key for the development of more effective BNCT in the future.

## 5. Conclusions

We synthesized novel KA-BSH. The basic research of BNCT has many limitations because it requires a neutron source, and this research also had its limitations. In such a situation, this preliminary study suggests that intravenous administration of KA-BSH could be a promising additional therapeutic modality, but further studies are warranted prior to using KA-BSH as a new boron compound for BNCT against malignant glioma. This is because there is almost no drug that exhibits an effect on the primary brain tumor by the intravenous administration of the novel boron compound.

## Figures and Tables

**Figure 1 cells-09-01551-f001:**
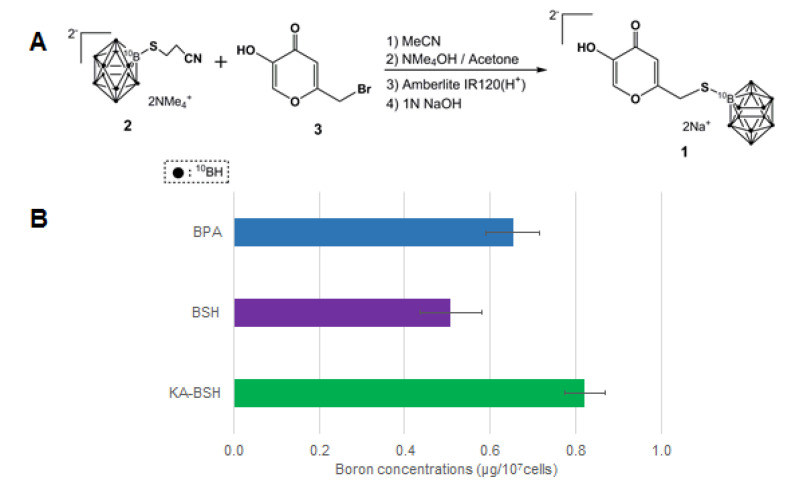
(**A**) The synthesis scheme of KA-BSH. (**B**) ^10^B concentration of F98 rat glioma cells 24 h after incubation in DMEM, including 1 mM of each boron compound. The ^10^B concentrations of tumor cells were significantly higher in the KA-BSH group than the BSH group (0.821 ± 0.047 vs. 0.508 ± 0.072 µg ^10^B/10^7^ cells; *p* = 0.0008) and BPA group (0.821 ± 0.047 vs. 0.654 ± 0.062 µg ^10^B/10^7^ cells; *p* = 0.0157).

**Figure 2 cells-09-01551-f002:**
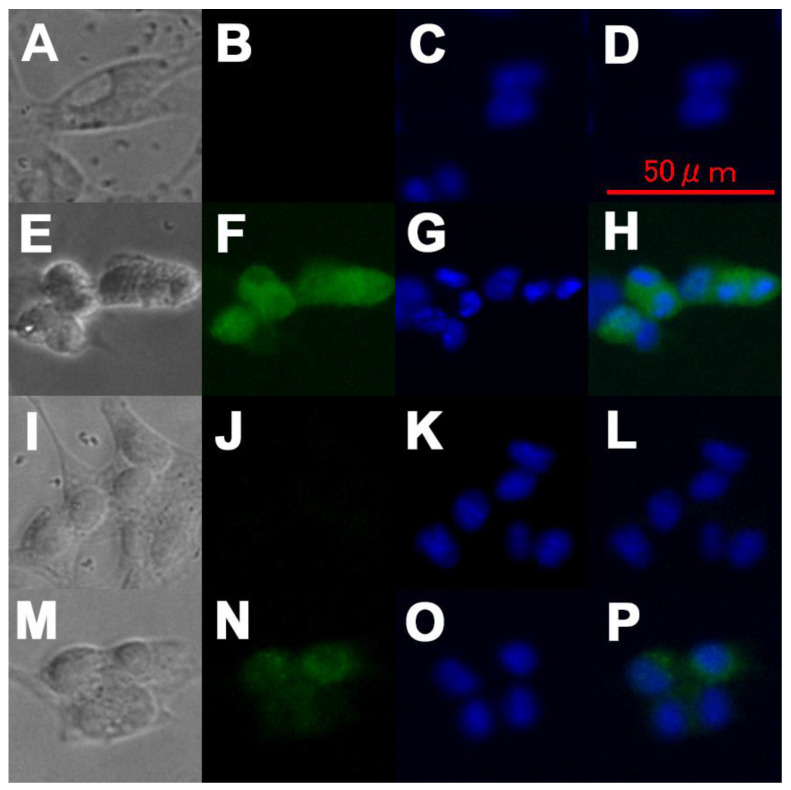
The micro-distribution of each boron compound in F98 rat glioma cells. (**A**,**E**,**I**,**M**) A phase-contrast micrograph of F98 cells that were cultured in DMEM (**A**), containing BPA (**E**), BSH (**I**), and KA-BSH (M). (**B**,**F**,**J**,**N**) A fluorescence micrograph of F98 cells that were cultured in DMEM (**B**), containing BPA (**F**), BSH (**J**), KA-BSH (**N**) stained with the anti-BSH antibody A9H3 (**B**,**J**,**N**), or anti-BPA antibody 2B10 (**F**). (**C**,**G**,**K**,**O**) A fluorescence micrograph of F98 cells that were cultured in DMEM (**C**), containing BPA (**G**), BSH (**K**), and KA-BSH (**O**) stained with Hoechst 33342. (**D**,**H**,**L**,**P**) Merged images. B and C (**D**), F and G (**H**), J and K (**L**), N and O (**P**).

**Figure 3 cells-09-01551-f003:**
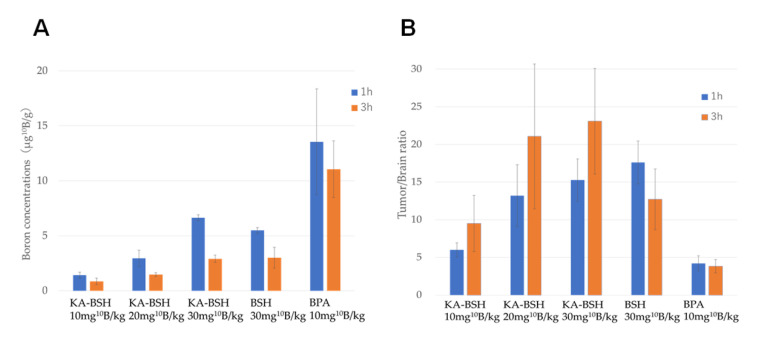
(**A**) Boron concentrations in a tumor in F98 glioma bearing rats, when we administrated BPA, BSH, and KA-BSH by i.v. (**B**) The B concentrations in the tumor, selected normal tissues of F98 glioma bearing rats as well as the tumor-to-normal brain (T/Br) ratio.

**Figure 4 cells-09-01551-f004:**
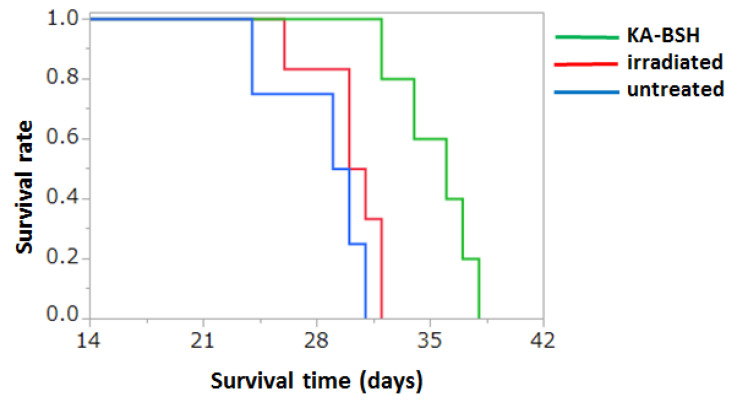
Kaplan–Meier survival curves for F98 glioma bearing rats following intravenous administration of KA-BSH followed by BNCT. Survival times in days after implantation have been plotted for untreated animals (blue line), irradiated controls (red line), and KA-BSH (green line). Median survival times of the KA-BSH group, neutron irradiation only group, and untreated group were 35.4 ± 8.0, 30.2 ± 2.2, and 28.5 ± 3.1 days, respectively. There were statistically significant differences between the KA-BSH group and neutron irradiation only group (*p* = 0.0053) and between the KA-BSH group and untreated group (*p* = 0.0027).

**Table 1 cells-09-01551-t001:** Summary of ^10^B concentrations, physical radiation doses, and biological radiation doses delivered in F98 glioma bearing rats.

Agent	Route	Time ^a^ (h)	^10^B Concentrations ± SD (μg ^10^B/g)						Physical Radiation Dose ^b^ (Gy)		Equivalent Dose ^c^ (Gy-eq)	
			Brain			Tumor			Brain	Tumor	Brain	Tumor
KA-BSH	iv	1	0.3	±	0.1	1.4	±	0.3	1.0	1.1	-	-
BPA	iv	1	3.0	±	0.8	16.0	±	4.0	1.3	3.0	2.2	9.4
Irradiated	-	-	0	±	0	0	±	0	1.0	1.0	-	-
Untreated	-	-	0	±	0	0	±	0	0	0	0	0

a: Time to euthanize rats after boron compound administration. b: Physical radiation dose estimates include contributions from gamma photons, ^14^N (n,p), ^14^C, ^10^B (n,α) ^7^Li, and ^1^H (n,n) ^1^H reactions. c: Equivalent dose is a value calculated using D_B_ × CBE_B_ + D_N_ × RBE_N_ + D_H_ × RBE_H_ + D_γ_; D_B_: Boron dose, D_N_: neutron dose, D_H_: hydrogen dose, D_γ_: gamma-ray dose; RBE_N_ is from RBE (Relative Biological Effectiveness) for D_N_, and this value is 3.0; RBE_H_ is from RBE (Relative Biological Effectiveness) for D_H_, and this value is 3.0; CBE_B_ is from CBE (Compound Biological Effectiveness) for D_B_, and in the case of BPA, this value is 3.8 for the tumor tissue and 0.9 for normal brain.

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
