# Peer review of "Synthesis and Evaluation of Dodecaboranethiol Containing Kojic Acid (KA-BSH) as a Novel Agent for Boron Neutron Capture Therapy"

_cells, 2020, doi:10.3390/cells9061551_

Round 1

Reviewer 1 Report

In this study, the authors developed a novel boron delivery agent containing kojic acid to use for boron neutron capture therapy in patients with glioma and described the in vitro and in vivo results that they have obtained for the characterization of this new compound.

The text is well organized and methods and results are fully described but there are some points that need to be clarified.

Major comments

Materials and methods

  1. Line 85-86, “Synthesis of S-(2-(5-hydroxy-4-oxo-4H-pyran-2-yl) methyl) thioundecahydro-closo- dodecaborate (2-) disodium salt (KA-BSH, 1)” is it a title or the verb is lacking?
  2. Line 101-104, can you add a reference about F98 cell line, please?

Results

  1. In figure 1B, Boron concentration is expressed as µg/107 cells whereas in the figure legend and in the text (lines 225-226) data are expressed in ng/106 cells, can authors uniform?
  2. Lines 224-225, authors have forgotten to indicate “in the KA-BSH group”, the sentence should be re-written in the Boron concentrations of F98 cells IN THE KA-BSH GROUP were significantly higher than after exposure to BSH……
  3. Paragraph 3.2, the Boron concentrations of F98 cells were significantly higher than after exposure to BSH and BPA but in B16 and in C6 cell lines (Fig.S1) the boron concentrations are higher in BPA group, authors have hypothesized why? The differences are significant?
  4. Paragraph 3.4, the cytotoxicity of KA-BSH is very low compared to BSH, why? The authors should discuss this in the discussion.
  5. Paragraph 3.5, in the text, the authors stated that figure 3 shows B concentrations in the tumor, selected normal tissues as well as tumor-to-normal brain ratio but figure 3 shows B concentration in tumor (3A) and the tumor/brain (3B). What is the difference between the ipsilateral and contralateral brain? Why did authors choose the ipsilateral brain as normal brain? F98 cells have been injected on one side so the contralateral brain should be healthy.

Please edit also the figure legend of figure 3.

Can you check T/Br values in table S1, please? Moreover, can you check T/Br values in table S1 with data in figure 3B? because in the table S1 T/Br value of KA-BSH 10 mg/kg 3h is 8.40 but in the graph the corresponding orange bar arrive to 6, or in the table S1 T/Br value of KA-BSH 20 mg/kg 3h is 10.25 in the graph the corresponding orange bar is more than 12, can you check all, please?

In figure 3B, add SD please.

Lines 262 and 263, data of the other organs are not shown, can you add a table with biodistribution data of all organs, please?

KA-BSH showed rapid clearance from normal tissues and it is a positive thing, authors should discuss better this point in the discussion.

  1. Why in the therapeutic experiments, KA-BSH has not been also compared to a BPA or BSH group?
  2. For therapeutic experiments the dose of 10 mg/kg has been chosen, why?
  3. In materials and methods, authors stated that for therapeutic experiments the rats were randomized into experimental groups of 6–8 animals (line 180) but in the table S2 the number of animals per group is 4-6, can authors verify?
  4. How is calculated %ILS?

Discussion

The authors should discuss in detail the advantages and disadvantages of the novel compound compared to BSH and BPA.

Minor comments

Line 119, A is a

Line 269, table 2 is table 1

Table S2 is not cited in the text.

Reviewer 2 Report

It was an absolute pleasure reading this manuscript and I can simply not find any fault here.

The authors are commended for a well thought out and executed experiment in a field desperately in need of new novel compounds to advance it's usefulness in the now competitive world of targeted therapy.

The manuscript is well written and the structure flows very well. The explanations of experiments are well supported by the conclusions without making unsubstantiated claims.

My reccommendation is for the manuscript to be published as is.

Author Response

Thank you very much for your reviewing and favorable remarks.

Reviewer 3 Report

The paper by Takeuchi et al. describes the synthesis and evaluation of dodecaboranethiol containing kojic acid (KA-BSH) as an agent for boron neutron capture therapy (BNCT) of melanoma and glioma cells in vitro and glioma in vivo. Malignant gliomas remain incurable, therefore there is an urgent need to develop an effective therapy for this disease. Undoubtedly, the results of these study may contribute to the improvement of BNCT. The experiments are well designed, the manuscript is logically structured, but there are some points which could improve this paper:

The major concerns are following:

  1. The authors legitimately highlighted that anticancer treatment should be selective and possibly not harmful to non-cancerous cell and tissues. Therefore, there is a question about the influence of KA-BSH on astrocytes or/and neurons. Biodistribution of KA-BSH amd its accumulation in the organs other than brain is promising, but there is no data on the cytotoxicity of KA-BSK on healthy brain cells. Such data (e.g. KA-BSH-treated astrocytes/neurons viability assay) would significantly improve the value of this manuscript.
  2. All cell lines used in this study are of rat- or mouse origin. While there are clearly a convinent model for mechanistic studies, they are still far from human cells; the studies on human cell lines either commercially available or (better) patient-derived (at least in the cytotoxic studies) would also improve the value of this paper.
  3. The selection of the origin of the cell lines is unobvious and should be justified. The Introduction section suggests that the idea of this study was to evaluate KA-BSH in glioma treatment. What was than the rationale of examination of its influence on melanoma cells?

Minor:

  1. Introduction, page 1. l. 40: the invasiveness of normal tissue by glioma cells is one of the main reasons of the theapeutic failure, but it is not the only one reason of glioma’s resistance to treatment – please re-write this sentence.
  2. The microscopic photos (Fig. 2) are not sharp, i sit possible to improve their quality?

Round 2

Reviewer 1 Report

The authors properly answered the comments. I have only another comment, the value of BPA 10 mg at 1 hour is 13.54 ± 4.82 as indicated in the text (line 266 and Table S1) but in figure 3A the corresponding blue bar is over 15, can you edit, please? 

Author Response

The authors properly answered the comments. I have only another comment, the value of BPA 10 mg at 1 hour is 13.54 ± 4.82 as indicated in the text (line 266 and Table S1) but in figure 3A the corresponding blue bar is over 15, can you edit, please?

Response : We used the "track change" function in Microsoft Word, which made it difficult for you to check the document. We checked and edited the figure 3.

Reviewer 3 Report

The manuscript by Takeuchi et al. is a revised version of the paper dealing with dodecaboranethiol containing kojic acid as a novel agent for boron neutron captured therapy.

The authors answered all the questions, but there are still some points in the manuscript that could be improved:

  1. I do recommend to include into the body of the manuscript the reason of examination of the influence of KA-BSH on melanoma cells as it is still not clear for the reader not very familiar with the topic of the article.
  2. I asked the authors to increase the resolution of the microscopic pictures. I assume that the new versions are underlined with a red line. Unfortunately, in my opinion the new (underlined in red) phase-contrast micrographs are much less sharp compared to the original versions. I therefore recommend to return to the original versions.

An editorial error: Fig. 2, letters A-D are on the separate page than the pictures; this could not appear in the final version (after all the fragments from the original version will be removed), but please check this point carefully.

Author Response

Point 1: I do recommend to include into the body of the manuscript the reason of examination of the influence of KA-BSH on melanoma cells as it is still not clear for the reader not very familiar with the topic of the article.

Response 1: We added the discussion about the reason of examination of the influence of KA-BSH on melanoma cells in line 302-304 (highlighted in yellow).

Point 2: I asked the authors to increase the resolution of the microscopic pictures. I assume that the new versions are underlined with a red line. Unfortunately, in my opinion the new (underlined in red) phase-contrast micrographs are much less sharp compared to the original versions. I therefore recommend to return to the original versions.

Response 2: Thank you for your advice. We returned the microscopic pictures to the original versions.

An editorial error: Fig. 2, letters A-D are on the separate page than the pictures; this could not appear in the final version (after all the fragments from the original version will be removed), but please check this point carefully.

We used the "track change" function in Microsoft Word, which made it difficult for you to check the document. We checked and edited Fig.2 in the final version.